# Acute-Phase Plasma Pigment Epithelium-Derived Factor Predicting Outcomes after Aneurysmal Subarachnoid Hemorrhage in the Elderly

**DOI:** 10.3390/ijms25031701

**Published:** 2024-01-30

**Authors:** Mai Nampei, Yume Suzuki, Hideki Nakajima, Hiroki Oinaka, Fumihiro Kawakita, Hidenori Suzuki

**Affiliations:** Department of Neurosurgery, Mie University Graduate School of Medicine, Tsu 514-8507, Japan; baske_05@yahoo.co.jp (M.N.); box_308044@yahoo.co.jp (Y.S.); zima0131@gmail.com (H.N.); hiros08290@gmail.com (H.O.); fxmx0216@yahoo.co.jp (F.K.)

**Keywords:** elderly patient, pigment epithelium-derived factor, prognostic factor, subarachnoid hemorrhage

## Abstract

Aneurysmal subarachnoid hemorrhage (SAH) has increased with the aging of the population, but the outcome for elderly SAH patients is very poor. Therefore, predicting the outcome is important for determining whether to pursue aggressive treatment. Pigment epithelium-derived factor (PEDF) is a matricellular protein that is induced in the brain, and the plasma levels could be used as a biomarker for the severity of metabolic diseases. This study investigated whether acute-phase plasma PEDF levels could predict outcomes after aneurysmal SAH in the elderly. Plasma samples and clinical variables were collected over 1–3 days, post-SAH, from 56 consecutive elderly SAH patients ≥75 years of age registered in nine regional stroke centers in Japan between September 2013 and December 2016. The samples and variables were analyzed in terms of 3-month outcomes. Acute-phase plasma PEDF levels were significantly elevated in patients with ultimately poor outcomes, and the cutoff value of 12.6 µg/mL differentiated 3-month outcomes with high sensitivity (75.6%) and specificity (80.0%). Acute-phase plasma PEDF levels of ≥12.6 µg/mL were an independent and possibly better predictor of poor outcome than previously reported clinical variables. Acute-phase plasma PEDF levels may serve as the first biomarker to predict 3-month outcomes and to select elderly SAH patients who should be actively treated.

## 1. Introduction

The aging of the population has led to an increase, in many developed countries, in elderly patients with subarachnoid hemorrhage (SAH) due to ruptured intracranial aneurysms. In Japan, the percentage of patients over 75 years of age exceeds 20% [1]. As endovascular therapy becomes more widespread and treatment techniques improve, the opportunities for aggressively treating elderly SAH patients are expanding [1,2,3]. However, previous studies have reported that poor outcomes clearly increase in SAH patients over 75 years of age [4,5], and, therefore, it is necessary to consider the targets separately.

Outcome determinants in aneurysmal SAH patients have been analyzed according to demographics; advanced age has been reported as one of the most important factors [6,7]. Elderly SAH patients are more frequently associated with pre-morbidities, comorbidities, poor admission World Federation of Neurological Surgeons (WFNS) grades, and more intracranial hematoma volume, resulting in poor outcomes [8]. In the elderly (≥75 years of age), however, further determinants for poor outcomes are limited to hypertension and increased intracranial hematoma volume in multivariate logistic regression analyses in our prospectively maintained SAH database at multiple institutions from 2013 to 2016 [8]. Thus, it is important to develop a biomarker to discriminate good and poor outcomes at an acute phase in individual elderly aneurysmal SAH patients; this may provide an important basis for making suggestions to the patient’s family regarding subsequent aggressive treatment.

Pigment epithelium-derived factor (PEDF) is a 50 kDa matricellular protein (MCP) with multifunctional properties [9]. PEDF was discovered as a neuronal differentiation activator in fetal retinal pigment epithelial cells, but is known to be induced in brain tissues [9,10]. MCP levels are generally higher in young people and decline with aging, but they increase in response to pathological conditions [10]. In addition, as MCPs are generally secreted into body fluids such as peripheral blood, MCPs, such as tenascin-C, periostin, osteopontin, and galectin, can be used to monitor the progression of inflammatory pathologies in ischemic and hemorrhagic strokes [10]. Understanding the role of MCPs has great potential in their use as prognostic biomarkers and therapeutic targets, but the information on MCPs in the central nervous system is limited [10]. PEDF has recently attracted a great deal of attention in terms of its metabolic regulation, and its circulation level has been reported to act as a biomarker for the severity of various metabolic diseases [11]: in healthy individuals aged 50 years and over, higher serum PEDF levels were observed in women, possibly reflecting their comorbidities and estrogen statuses [12]. It is well known that aging induces a shift toward proinflammatory phenotypes in the brain and the periphery, as well as blood–brain barrier (BBB) dysfunction, and that older women are more likely to have strokes leading to worse outcomes, although women are protected from stroke before menopause [13]. In animal models of cerebral ischemia, in addition, endogenous PEDF was reactively upregulated in injured brain tissues [14,15,16]. Thus, we hypothesized that circulating PEDF levels are elevated in elderly SAH patients in a manner that reflects their comorbidities, complications, and severity of brain injury and that their PEDF levels are useful as prognostic biomarkers. However, the clinical significance of plasma PEDF levels has never been investigated in SAH patients.

The aim of this study was to clarify the association between plasma PEDF levels at an acute phase and 3-month outcomes in SAH patients aged ≥75 years and to determine whether the PEDF levels are useful as a prognostic biomarker in elderly aneurysmal SAH patients. If acute-phase plasma PEDF levels can serve as a biomarker for predicting 3-month outcomes in elderly SAH patients, it would become the first clinical index to allow us to select elderly SAH patients who can achieve good outcomes with active treatment.

## 2. Results

### 2.1. Clinical Characteristics of Elderly SAH Patients According to 3-Month Outcome

In the Prospective Registry for Searching Mediators of Neurovascular Events After Aneurysmal SAH (pSEED) [17,18,19], 275 consecutive aneurysmal SAH patients were registered. The inclusion criteria for this study were ≥75 years of age at onset, pre-onset modified Rankin Scale (mRS) score 0–2, obliteration of ruptured intracranial aneurysms within 48 h of onset, and plasma sampling at days 1–3 post-SAH. Patients with infectious diseases that may increase plasma PEDF levels were excluded. Thus, 56 elderly SAH patients (≥75 years) were eligible for this study (Figure 1). For four patients whose 3-month outcomes were unknown, the outcome at discharge was used instead; their outcomes were mRS 0, 2, 4, and 5, respectively. In addition, angiographic vasospasm was not evaluated in one patient due to poor general condition; this patient was included in the study and analyzed excluding the missing data.

The clinical variables in 56 elderly SAH patients (≥75 years) with good and poor 3-month outcomes are shown in Table 1. Of the 56 patients, 41 (73.2%) had poor outcomes (mRS 3–6), and 8 patients (14.3%) died. The median age of the 56 patients was 81 years (interquartile range, 79.0–83.0), and the population consisted of 48 female patients (85.7%), 33 patients (58.9%) with admission WFNS grades IV–V, 30 patients (53.6%) with modified Fisher grade 4, and 30 patients (53.6%) with acute hydrocephalus. Cerebrospinal fluid (CSF) drainage was performed in 17 patients (30.4%). Ruptured aneurysms were obliterated with clipping in most patients (42 patients, 75.0%) or with simple coiling in the other patients. Treatment complications were observed in 16 patients (28.6%); 14 of these had cerebral infarction, one patient had cerebral hemorrhage, and one patient had cerebral contusion. Angiographic vasospasm occurred in 13 patients (23.2%), delayed cerebral ischemia (DCI) occurred in 8 patients (14.3%), and delayed cerebral infarction occurred in 19 patients (33.9%). In total, 23 patients (41.1%) underwent ventriculoperitoneal or lumbo-peritoneal CSF shunting for chronic hydrocephalus (chronic shunt-dependent hydrocephalus (CSDH)).

As for baseline demographic and clinical variables, patients with poor 3-month outcomes were associated with worse modified Fisher computed tomography (CT) grades, although other factors, such as age, sex, comorbidities, smoking, admission WFNS grades, acute hydrocephalus, and ruptured aneurysm location, were not different between patients with good and poor outcomes (Table 1). None of the treatment-related factors were different between patients with good and poor outcomes, including aneurysmal treatment modalities (clipping or coiling), CSF drainages, treatment complications, or prophylactic medications for DCI (Table 1). The incidences of DCI, delayed cerebral infarction diagnosed using CT scans, and CSDH were higher in patients with poor outcomes compared to those with good outcomes, but the difference did not reach statistical significance. In contrast, angiographic vasospasm occurred similarly in both outcome groups.

### 2.2. Acute-Phase Plasma PEDF Concentrations in Elderly SAH Patients

In the 56 elderly SAH patients, the PEDF levels in the stocked plasma samples obtained at days 1–3 after SAH onset were determined using a commercially available enzyme-linked immunosorbent assay kit. The plasma PEDF levels in an acute phase in the 56 elderly SAH patients were significantly higher when compared with the plasma PEDF levels in 10 patients with unruptured intracranial aneurysms (five males and five females; mean age, 66.7 ± 8.9 years; Figure 2A). Because unruptured aneurysms in the elderly were not treated and, therefore, their plasmas were not obtained, the age in the unruptured aneurysm group was significantly younger than that in the SAH group (81.8 ± 4.4 years; *p* < 0.001, Mann–Whitney U test), but there were no gender differences in the plasma PEDF concentrations in either the SAH group or the unruptured aneurysm group (male vs. female = 17.3 ± 4.8 vs. 14.1 ± 4.9µg/mL, *p* = 0.086 and 9.9 ± 2.5 vs. 7.8 ± 0.9µg/mL, *p* = 0.110, respectively; unpaired *t*-test).

When the acute-phase plasma PEDF levels were compared between the elderly SAH patients who ultimately had good and poor outcomes, the PEDF levels in patients with ultimately poor 3-month outcomes were significantly higher than those in the good-outcome group (15.4 ± 4.6 vs. 12.2 ± 5.4 µg/mL, respectively; Figure 2B).

### 2.3. Independent Determinants for Poor Outcome in Elderly SAH Patients

When a receiver-operating characteristics (ROC) curve was constructed for acute-phase plasma PEDF levels to predict poor 3-month outcome, the area under the curve was 0.730 and the cutoff value was determined as the plasma PEDF levels at days 1–3 ≥12.6 μg/mL using the Youden index, with a sensitivity of 75.6% and a specificity of 80.0% (Figure 3). The multicollinearity among the PEDF cutoff values and clinical factors that were obtained at days 1–3 post-SAH was evaluated using variance inflation factor (VIF). As a VIF > 5 of multicollinearity was observed between CSF drainage (VIF = 22.189) and ventricular drainage (VIF = 20.739), ventricular drainage with a lesser *p*-value on univariate analyses (Table 1) was retained, and CSF drainage was removed. VIF was calculated again to confirm that there was no multicollinearity. An evaluation using Pearson’s correlation coefficient was also performed to confirm that there was no correlation between any of the variables, and then multivariate analyses were performed. The multivariate analyses revealed that advanced age (adjusted odds ratio (aOR), 1.226; 95.0% confidence interval [CI], 1.002–1.500; *p* = 0.048), modified Fisher grade (aOR, 3.944; 95.0% CI, 1.196–13.009; *p* = 0.024), and acute-phase plasma PEDF levels ≥12.6 µg/mL (aOR, 21.493; 95.0% CI, 2.913–158.553; *p* = 0.003) were independent factors for poor 3-month outcomes, while the use of a statin drug (aOR, 0.089; 95.0% CI, 0.011–0.726; *p* = 0.024) was an independent factor for good 3-month outcomes (Table 2). Admission WFNS grade did not remain as an effective factor: when an ROC curve was created for admission WFNS grade to predict poor outcomes, the area under the curve was 0.563.

## 3. Discussion

Elderly SAH patients have a high rate of poor clinical grades on admission, causing poor outcomes [5,8,20]. Although the definition of elderly SAH patients differs among articles, previous studies reported that poor outcomes especially increase in patients over 75 years of age [4,5], i.e., the target of this study. The indications of aggressive treatment for elderly SAH patients have expanded [1,2,3] and are expected to further increase in the future. Therefore, it is useful to predict outcomes at an acute stage when considering treatment indications for elderly SAH patients. Several studies have reported possible determinant factors, mainly based on demographics, for poor outcomes in elderly SAH patients; these factors included more advanced age, male sex, history of hypertension, worse WFNS or other clinical grades, higher modified Fisher grade, larger intracerebral or intraventricular hematoma volume, acute hydrocephalus, and more severe angiographic vasospasm [1,4,6,7,8,21,22]. However, most of the determinants are not valid when predicting the outcome of SAH patients limited to the elderly, because the determinants themselves are the characteristics of elderly SAH patients, who often have poor general health as well as intracranial conditions on admission [8]. Thus, outcome predictors that can be adapted to and are, ideally, specific to elderly SAH patients are needed. The present study first revealed that elevated plasma PEDF levels at days 1–3 post-SAH were an independent predictor of poor outcomes in elderly SAH patients over 75 years of age and that using ROC curve analyses (area under the curve, 0.730) and plasma PEDF levels at days 1–3 ≥12.6 μg/mL differentiated 3-month outcomes with high sensitivity (75.6%) and specificity (80.0%). In this study, acute-phase plasma PEDF levels had a higher aOR than age and modified Fisher grade, which assess both SAH and intraventricular hematoma volume [23]. In contrast, admission WFNS grades and sex were not independent determinants for poor outcomes in this study. Taken together, the findings suggest that acute-phase plasma PEDF levels are more useful to predict outcomes for elderly SAH patients compared with previously reported outcome determinants such as age and admission neurological status. Setting a specific cutoff value of plasma PEDF may facilitate the prediction of outcome in the acute phase of SAH and allow medical resources to be concentrated on elderly patients who are expected to have a good outcome.

### 3.1. MCPs as a Biomarker

In recent years, some MCPs were reported to increase in peripheral blood after aneurysmal rupture and attention has been paid to MCP as a biomarker for predicting delayed-onset neurovascular events and final outcomes after aneurysmal SAH [17,19,24,25,26]. MCPs are characterized as molecules easily moving or secreted from injured sites into various body fluids such as circulating blood and CSF, although the mechanism of the moving or secretion remains unclear [27]. Although it is not certain whether some kinds of MCPs are upregulated in post-SAH-injured brains and then secreted into peripheral blood in a clinical setting, our previous clinical study suggested that MCP periostin in CSF leaked out into intravascular circulating blood through a disrupted BBB to predict the subsequent development of DCI after aneurysmal SAH [18]. Reportedly, higher admission plasma levels of another MCP, thrombospondin-1, were an independent factor for poor 6-month outcome and had similar predictive performance, regarding outcomes, to WFNS grades and Fisher scores in aneurysmal SAH patients [24]. Elevated plasma osteopontin levels within 3 days of aneurysmal SAH onset were also an independent factor predicting poor 3-month outcome [25]. Higher plasma levels of galectin-3 at admission or by 3 days post-SAH onset were associated with poor 3- or 6-month outcomes in clinically mild cases or regardless of the clinical severity [17,26]. Plasma levels of fibulin-5 were increased at days 4–6 post-SAH, leading to poor 3-month outcomes [19]. Osteopontin, galectin-3, and fibulin-5 are all MCPs. However, all of the analyses included patients of all ages, and it remains unclear whether MCP levels can be predictive of outcome when restricted to elderly SAH patients. As for PEDF, it is also an MCP, but its significance in SAH has never been investigated clinically or experimentally. To the best of the authors’ knowledge, this is the first study showing that plasma concentrations of MCP PEDF can serve as a biomarker for predicting outcomes in elderly SAH patients.

### 3.2. PEDF and Early Brain Injury (EBI)

PEDF is an extracellular matrix protein and MCP, and belongs to the Serpin family [28]. MCPs are generally not highly expressed at a steady state in adult tissues, but their expression easily increases with a variety of phenotypes in response to pathological conditions and diverse injuries [29,30]. PEDF is increased by hyperosmotic stress in cultured human corneal epithelial cells [31] and is upregulated, at least, in pericytes in mouse models of middle cerebral artery (MCA) occlusion and cold injury [16,32]. In addition, PEDF upregulation was observed in astrocytes after MCA occlusion in rats [14]. Thus, endogenous PEDF is expected to be induced upon ischemic tissue damage in the brain [14,16], but it has not been investigated whether PEDF is upregulated in the brain tissue after experimental or clinical SAH. Post-SAH brain injuries are divided into two types of injuries depending on the time of occurrence: injuries in which EBI occurs within 72 h of SAH onset and injuries in which DCI develops after 72 h following SAH onset. Each of these types of brain injuries have different mechanisms [33]. EBI is triggered by the rupture of an intracranial aneurysm followed by increased intracranial pressure—causing transient global cerebral ischemia—as well as mechanical brain injuries, and its main pathologies include the hypoxic disturbance of metabolisms, neuroinflammation, microthrombosis, BBB disruption, neuronal apoptosis, seizure, and cortical spreading depolarization [34,35,36]. Thus, PEDF may be upregulated in brain tissues by ischemic injuries associated with EBI [14,16]. The severity of EBI is known to be correlated with admission WFNS grades [34], and more severe EBI is more likely to lead to angiographic vasospasm-related DCI or vasospasm-unrelated DCI and poor outcomes [30,33,37]. In this study, because angiographic vasospasm was not associated with outcome, it is suggested that EBI itself or DCI unrelated to vasospasm may be more important for poor outcome in elderly SAH patients. Thus, it has been proposed that EBI is the most important prognostic factor in aneurysmal SAH [38,39,40]; this is consistent with the results of our study targeting the elderly. Although this study showed that higher plasma PEDF levels at days 1–3 post-SAH were associated with worse outcomes in elderly SAH patients over 75 years of age, it is unknown whether an increase in PEDF in peripheral blood originates from the brain. However, if PEDF levels in the peripheral blood increase to reflect the severity of EBI in an acute stage of SAH, it seems logical that acute-phase plasma PEDF levels would be an important prognostic factor. This hypothesis is also consistent with the findings in our cohort (pSEED) that worse admission WFNS grades caused higher plasma PEDF levels in a study of SAH patients of all ages.

### 3.3. Possible Significance of Elevated Acute-Phase Plasma PEDF Levels

Given the above results, PEDF appears to be harmful to the brain function after aneurysmal SAH, especially in the elderly. However, PEDF is a multifunctional MCP and is known to act neuroprotectively in many situations [9,41]. In a rat model of transient MCA occlusion, overexpressed PEDF or transvenously administered PEDF protected neurons and other cells from ischemic insult, leading to a reduction in cerebral infarction, BBB permeability, brain edema formation, and neuroinflammation [14,15]. The intracerebroventricular administration of recombinant PEDF also decreased BBB disruption, brain edema, and neuronal cell death in a mouse model of transient MCA occlusion [16]. In addition, intraperitoneal administration of PEDF inhibited brain edema formation after cold injury by blocking vascular endothelial-growth-factor-mediated signaling in mice [32]. In summary, basic research has shown that PEDF is a neurotrophic molecule with neuroprotective and anti-permeability effects, at least in ischemic and other brain injuries. Because transient ischemia is an important inducer of EBI, and because neuronal apoptosis, BBB disruption and brain edema formation are the main components of EBI, as mentioned above [34,36], the authors consider that PEDF may also exert neuroprotective effects in the brain after aneurysmal SAH. In this study, although elevated acute-phase plasma PEDF levels were independent predictors of poor 3-month outcomes, the following explanation could be possible: (1) plasma PEDF may increase, reflecting the severity of precedingly developed EBI, and may indicate the degree of EBI more precisely than admission WFNS grades; (2) upregulated PEDF may have neuroprotective effects against EBI, but the endogenous PEDF levels may be neither enough to recover the proceeding EBI nor to prevent EBI from progressing to DCI; and (3) as a result, higher acute-phase plasma PEDF levels may be closely linked to poor outcomes after aneurysmal SAH. The same phenomenon has been observed with another MCP, osteopontin, which has been repeatedly demonstrated to be neuroprotective against brain injuries after experimental SAH [42,43,44]; however, plasma osteopontin levels at days 1–12 post-SAH were higher in patients who ultimately had a poor outcome [25]. Thus, our hypotheses described above may be reasonable.

### 3.4. Limitations of This Study

This study has several limitations. First, the study population is relatively small because the target population was limited to elderly subjects ≥75 years of age. The findings need to be confirmed in a validation cohort. Second, because the study excluded patients with obliteration of ruptured aneurysms >48 h post-SAH, patients with infection that may increase plasma PEDF levels, and patients without a plasma sample, it cannot be denied that the results of this study were biased. Third, it was not tested whether PEDF is superior to previously reported biomarkers. This is because there are no established or validated biomarkers known in SAH, although many plausible biomarkers have been reported, including MCPs and non-MCPs [45]. Fourth, the exact mechanisms by which plasma PEDF increases and how PEDF works, including the site of PEDF production, are unknown in aneurysmal SAH. However, this is the first study to show acute-phase plasma PEDF levels as an independent predictor of 3-month clinical outcomes in elderly SAH patients. Elevated plasma PEDF levels at days 1–3 after onset may be a better predictor of poor outcomes in elderly SAH patients than the clinical factors that have been reported to be outcome determinants, although further basic and clinical investigations are required to clarify the function, mechanisms, and clinical significance of PEDF.

## 4. Materials and Methods

All procedures performed in the studies involving human participants were carried out in accordance with the ethical standards of the institutional and/or national research committee and with the 1964 Declaration of Helsinki and its later amendments or comparable ethical standards. The study was approved by the ethical committee of Mie University Hospital (approval numbers 2544 and H2018-031), and written informed consent was obtained from the relatives.

### 4.1. Study Population

The present study used the clinical data and plasma samples collected in the pSEED that was conducted in 9 stroke centers in Mie prefecture in Japan between September 2013 and December 2016 (listed in Appendix A) [17,18,19]. The inclusion criteria were as follows: ≥20 years of age at onset, pre-onset mRS 0–2, SAH diagnosed using CT scans, saccular aneurysm as the cause of SAH confirmed on CT angiography or digital subtraction angiography, and aneurysmal obliteration via surgical clipping or endovascular coiling. The following patients were excluded, following the recommendations of previous studies [17,18,19], because their pathology is different from SAH caused by a ruptured saccular aneurysm: patients with dissecting, traumatic, mycotic, and arteriovenous malformation-related aneurysms or SAH of unknown etiology. After angiographic confirmation of a ruptured intracranial aneurysm, surgical clipping or endovascular coiling of the lesion was performed as judged by the attending neurosurgeon to be appropriate for the individual patient. From the registered 275 consecutive SAH patients, cases with obliteration of ruptured aneurysms >48 h post-SAH (*n* = 12), infection that may increase plasma PEDF levels (*n* = 33), missing plasma sample (*n* = 9), and <75 years of age at onset (*n* = 165) were excluded. Finally, 56 elderly SAH patients (≥75 years) were retrospectively analyzed to clarify the association between acute-phase plasma PEDF levels and 3-month outcomes (Figure 1).

### 4.2. Clinical Variables

The variables included age, sex, comorbidities, social and family histories, mRS before onset, admission WFNS grades, modified Fisher grade [23] on admission CT scans, acute hydrocephalus, ruptured aneurysm location (anterior or posterior circulation), CSF drainage (ventricular, cisternal, or spinal drainage), treatment modalities (clipping or coiling), procedural complications (cerebral infarction, hemorrhage, contusion, and others), prophylactic medications for DCI (intravenous injections of fasudil hydrochloride; oral or enteral administration of cilostazol; eicosapentaenoic acid; and statin), DCI, angiographic vasospasm, delayed cerebral infarction, CSDH, and outcomes. The timing of treatment, treatment modality used for aneurysmal obliteration, and other medical management or treatment strategies were decided by the onsite treating neurosurgeons and were not limited. As for CSF drainage, a ventricular catheter was placed to manage acute hydrocephalus in all patients. Ventricular drainage was also placed to control brain swelling or increased intracranial pressure, and cisternal drainage was placed in the basal cistern to promote SAH clearance after surgical clipping in some cases, according to the preference of treating neurosurgeons. Lumbar spinal drainage was placed, irrespective of clipping or coiling, to promote SAH clearance and/or to manage progressive ventriculomegaly that occurred postoperatively and within 14 days of onset.

Acute hydrocephalus was diagnosed via ventriculomegaly on admission CT scans and was considered to cause disturbance of consciousness. Hemorrhagic or ischemic complications related to clipping or coiling were diagnosed on CT or magnetic resonance (MR) images on the first post-operative or post-intervention day. DCI was defined as otherwise unexplained clinical deterioration (i.e., focal neurological impairments, a decrease of at least two points on the Glasgow Coma Scale, or both) that lasted for at least one hour [46]; other potential causes of clinical deterioration were rigorously excluded. Angiographic vasospasm was defined as ≥50% narrowing compared with the baseline vessel diameter of major cerebral arteries on CT, MR, or digital subtraction angiographies regardless of clinical symptoms. Delayed cerebral infarction was defined as a newly developed cerebral infarct on CT scans that was undetected on the images one day after surgery or intervention. CSDH was diagnosed based on the following 2 findings: (1) no detectable causes of persistent conscious disturbance or neurological deterioration other than hydrocephalus that occurred after day 14 post-SAH and (2) progressively increased ventricular size with an Evans index of ≥0.30. CSDH was treated with CSF shunting. Outcomes were evaluated at discharge and 3 months post-SAH: 3-month mRS 0–2 was defined as good outcome and 3–6 was defined as poor outcome. These events were assessed and determined at each stroke center, and the organizing committee qualified them.

### 4.3. Measurement of Plasma PEDF

Blood samples were collected with minimal stasis from peripheral veins early in the morning at days 1–3 post-SAH following aneurysmal obliteration. All blood samples were immediately centrifuged for 5 min at 3000 rpm to separate cellular materials from the supernatant plasma, and the plasma samples were stored at −78 °C until assayed. Experienced technicians unaware of the clinical information determined plasma PEDF levels, which were quantified using a commercially available enzyme-linked immunosorbent assay kit for human PEDF (RD191114200R; BioVendor, Brno, Czech Republic).

As a control, plasma samples were collected from 10 patients with unruptured intracranial aneurysms and no concomitant diseases that potentially affect PEDF expression levels. Written informed consent was provided prior to any invasive procedure. Plasma PEDF levels in control samples were determined as described above.

### 4.4. Statistical Analyses

Continuous variables were expressed as mean ± standard deviation and standard error of the mean for graphs or as median ±25th to 75th percentiles. Comparisons of continuous variables between two groups were performed using an unpaired t-test if each population followed a normal distribution using the Shapiro–Wilk test; otherwise, this was carried out using the Mann–Whitney U test. Categorical variables were presented as a frequency or percentage and compared using Pearson’s chi-square test or Fisher’s exact test, as appropriate.

To minimize the potential bias introduced by choosing a single cutoff value, ROC curve analyses were performed to determine the area under the curve and the best cutoff value of plasma PEDF levels for poor outcome using the Youden index. The sensitivity and specificity were determined for the cutoff value of a plasma concentration of PEDF to predict poor 3-month outcome. For the following multivariate analyses, multicollinearity among all variables was evaluated using VIF, and factors with VIF >5 were excluded except for a factor with the smallest *p*-value on univariate analyses exploring poor outcome-related variables. All variables used in the multivariate analyses were also evaluated using Pearson’s correlation coefficient: r > 0.5 of correlation was judged significant, and only the variable with the smallest *p*-value was used as a candidate variable among similar clinical variables that were intercorrelated. Multivariate logistic regression analyses were performed using a forward stepwise method with 3-month dichotomous mRS outcome (good or poor) as the dependent variable and candidate variables selected, as described above, as independent variables. aORs with 95.0% CIs were calculated, and the independence of variables was tested using the likelihood ratio test on reduced models. A *p*-value of <0.05 was considered significant. IBM SPSS Statistics version 28.0.0.0 (IBM, Armonk, NY, USA) was used for all statistical analyses.

## Figures and Tables

**Figure 1 ijms-25-01701-f001:**
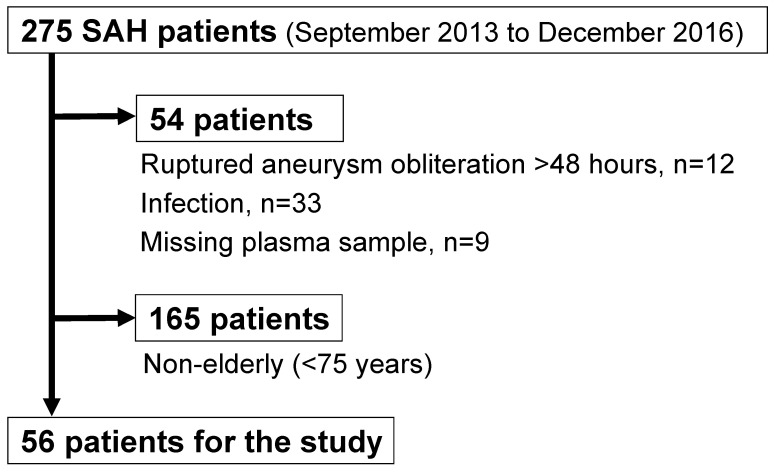
Flow chart indicating the included and excluded patients in this study. SAH, subarachnoid hemorrhage.

**Figure 2 ijms-25-01701-f002:**
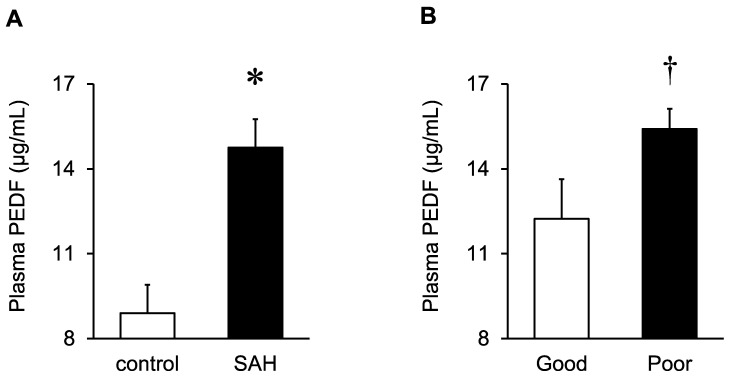
Comparisons of plasma pigment epithelium-derived factor (PEDF) levels between 10 control patients with unruptured cerebral aneurysms and 56 elderly subarachnoid hemorrhage (SAH) patients (**A**) and between elderly SAH patients with ultimately good and poor 3-month outcomes (15 and 41 patients, respectively; (**B**)). Plasma samples were collected at days 1−3 post-SAH from elderly SAH patients. Data are expressed as mean ± standard error of the mean. * *p* < 0.01 versus control (unpaired *t*-test; (**A**)), † *p* < 0.05 versus good outcome (Mann–Whitney U test; (**B**)).

**Figure 3 ijms-25-01701-f003:**
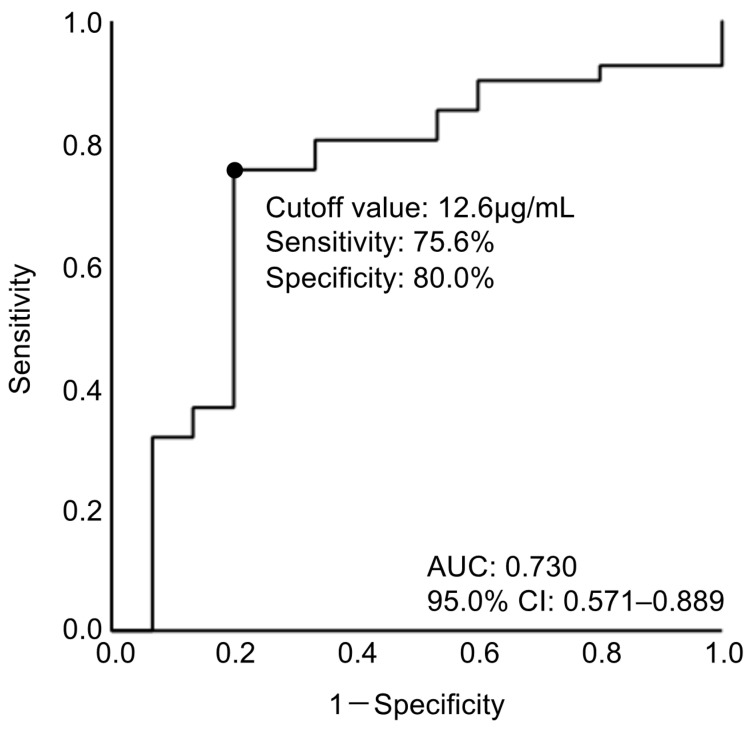
Receiver-operating characteristic curve for plasma PEDF levels at days 1–3 after subarachnoid hemorrhage (SAH) according to good and poor 3-month outcomes in elderly SAH patients. AUC, area under the curve; CI, confidence interval.

**Table 1 ijms-25-01701-t001:** Characteristics of elderly subarachnoid hemorrhage (SAH) patients with good and poor outcomes according to 3-month modified Rankin scale (mRS).

	Good (mRS 0–2) *n* = 15	Poor (mRS 3–6) *n* = 41	*p*-Value
Age, median (±25th–75th percentiles), years	80.0 (78.0–83.0)	81.0 (79.5–84.0)	0.468 ^a^
Sex, female	14 (93.3)	34 (82.9)	0.428 ^c^
Comorbidities			
Hypertension	6 (40.0)	22 (53.7)	0.365 ^b^
Diabetes mellitus	2 (13.3)	7 (17.1)	0.602 ^c^
Dyslipidemia	2 (13.3)	8 (19.5)	0.713 ^c^
Current smoking	2 (13.3)	3 (7.3)	0.602 ^c^
Family history of SAH	0	5 (12.2)	0.309 ^c^
Pre-onset mRS			0.349 ^b^
0	13 (86.7)	30 (73.2)	
1	2 (13.3)	6 (14.6)	
2	0	5 (12.2)	
Admission WFNS grade			0.938 ^b^
I	2 (13.3)	4 (9.8)	
II	4 (26.7)	10 (24.4)	
III	1 (6.7)	2 (4.9)	
IV	5 (33.3)	12 (29.3)	
V	3 (20.0)	13 (31.7)	
Modified Fisher grade			0.012 ^b^
1	3 (20.0)	2 (4.9)	
2	0	1 (2.4)	
3	9 (60.0)	11 (26.8)	
4	3 (20.0)	27 (65.9)	
Acute hydrocephalus	7 (46.7)	23 (56.1)	0.531 ^b^
Ruptured AN location			
Anterior circulation	14 (93.3)	38 (92.7)	1.000 ^c^
Posterior circulation	1 (7.7)	3 (7.3)	
Cerebrospinal fluid drainage	4 (26.7)	13 (31.7)	1.000 ^c^
Ventricular	3 (20.0)	13 (31.7)	0.513 ^c^
Cisternal	0	1 (2.4)	1.000 ^c^
Spinal	3 (20.0)	1 (2.4)	0.055 ^c^
Treatment modality			
Clipping	9 (60.0)	33 (80.5)	0.165 ^c^
Coiling	6 (40.0)	8 (19.5)	
Procedural complication	5 (33.3)	11 (26.8)	0.741 ^c^
Prevention of DCI			
Fasudil hydrochloride	14 (93.3)	35 (85.4)	0.661 ^c^
Cilostazol	14 (92.3)	32 (78.0)	0.259 ^c^
EPA	6 (40.0)	13 (31.7)	0.562 ^b^
Statin	7 (46.7)	9 (22.0)	0.097 ^c^
Angiographic vasospasm	4 (26.7)	9 (22.0)	0.730 ^c^
DCI	0	8 (19.5)	0.093 ^c^
Delayed cerebral infarction	4 (26.7)	15 (36.6)	0.488 ^b^
CSDH	4 (26.7)	19 (46.3)	0.185 ^b^

Data, the number of patients (% of all patients per group) unless otherwise specified. *p*-values, ^a^ Mann–Whitney *U* test, ^b^ Pearson’s chi-square test or ^c^ Fisher’s exact test. Only modified Fisher computed tomography grades are significantly different between good and poor outcomes (*p* < 0.05). AN, aneurysm; CSDH, chronic shunt-dependent hydrocephalus; DCI, delayed cerebral ischemia; EPA, eicosapentaenoic acid; WFNS, World Federation of Neurological Surgeons.

**Table 2 ijms-25-01701-t002:** Multivariate logistic regression analyses for poor 3-month outcomes in elderly patients with subarachnoid hemorrhage.

	Multivariate Analysis		
	aOR	95.0% CI	*p*-Value
Age	1.226	1.002–1.500	0.048
Modified Fisher grade	3.944	1.196–13.009	0.024
Use of statin drug	0.089	0.011–0.726	0.024
PEDF ≥ 12.6 µg/mL	21.493	2.913–158.553	0.003

Multivariate logistic regression analyses are conducted using the forward selection method. Plasma pigment epithelium-derived factor (PEDF) levels at days 1–3 are categorized using the cutoff value that was determined in Figure 3. aOR, adjusted odds ratio; CI, confidence interval.

## Data Availability

The data from this study will be made available to qualified investigators upon reasonable inquiry.

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
