# Peer review of "Acute-Phase Plasma Pigment Epithelium-Derived Factor Predicting Outcomes after Aneurysmal Subarachnoid Hemorrhage in the Elderly"

_ijms, 2024, doi:10.3390/ijms25031701_

Round 1
Reviewer 1 Report
Comments and Suggestions for Authors
The aim of the paper "Acute-phase plasma pigment epithelium-derived factor predicting outcomes after aneurysmal subarachnoid hemorrhage in the elderly" is to investigate if measurements of plasma PEDF levels in an acute phase are useful to predict outcomes after aneurysmal SAH in the elderly.
1. A native English speaker should check the entire Manuscript.
2. Title: The title is adequate.
3. Abstract: The abstract is generally clear and well-structured. It provides background information, outlines the methods, presents the results, and concludes with key findings. Nonetheless, some issues need to be addressed. For example, the sentence „The opportunities to treat elderly patients with aneurysmal subarachnoid hemorrhage (SAH) are increasing with the aging of the population, but the outcome of elderly SAH patients is very poor“ – is not well constructed, SAH is increasing due to the aging, that is primarily, not the opportunities to treat person, those come second. There is no explanation of what PEDF is. The abstract mentions a sample size of 56 elderly SAH patients ≥75 years, but it lacks information on how the sample was selected or whether it represents a diverse population. There is no discussion of the results.
4. Introduction: The biggest problem is the English language since the whole document is confusingly written and is hard to understand.
The introduction covers a broad spectrum of topics, from demographic trends to the potential biomarker. Ensuring cohesiveness and the progression of ideas can enhance overall readability.
„The aging of the population in many countries has led to an increase in elderly patients with subarachnoid hemorrhage (SAH) due to ruptured intracranial aneurysms, with the percentage of the patients over 75 years of age exceeding 20% [1].“- is this characteristic in all countries or many countries?
„In addition, as MCPs are generally secreted into body fluids such as peripheral blood, MCPs can be used to monitor the progression of some kinds of brain diseases [10]. “ – what kind of brain diseases?
When mentioning previous studies on poor outcomes in SAH patients over 75 years of age, it would be beneficial to briefly highlight key findings or methodologies of those studies to provide context.
Please explicitly state the significance of investigating PEDF in elderly SAH patients. How would the identification of PEDF as a prognostic biomarker contribute to current knowledge or clinical practice?
Also, please explicitly state the research gap or the need for the current study. What specific knowledge or understanding is lacking in the literature that this study aims to address?
5. The Materials & Methods: This section is comprehensive, but its readability could be improved by breaking down complex sentences and concepts, particularly when explaining procedures and criteria.
„The following patients were excluded: patients with dissecting, traumatic, mycotic and arteriovenous malformation-related aneurysms or SAH of unknown etiology.“-the authors should mention why they were excluded.
The criteria for excluding cases (e.g., obliteration of ruptured aneurysms > 48 hours post-SAH, infection, missing plasma sample) are mentioned, but there's a lack of discussion about how these exclusions might impact the representativeness of the sample.
The control samples - the section lacks details on how these control subjects were selected and whether they were appropriately matched.
6. Results: Table 1 provides p-values without clearly stating the significance level (p < 0.05). It's crucial to define the threshold for significance. The clarity of data presentation could be improved. Consider simplifying the table or providing subheadings to enhance readability. Ensure consistency in reporting percentages; for example, some percentages are reported with decimal points, while others are whole numbers. Is the determined cutoff value consistent with or differs from values reported in the literature? Discuss any variations and their potential implications. How selecting a particular cutoff value might impact clinical decision-making.
7. Discussion: Improve organization and structure to enhance clarity. Consider breaking down the discussion into subsections to address different aspects sequentially. While the discussion mentions that plasma PEDF levels are an independent predictor of poor outcomes in elderly SAH patients, there could be a more in-depth exploration of the clinical implications. For example, discuss how this information could impact patient management or guide treatment decisions in elderly SAH cases. You can further compare the study findings with existing literature on biomarkers in SAH. Address how the results align or differ from previous studies, especially those investigating different MCPs, and highlight the novelty or significance of the current study. While the limitations are mentioned, it would be beneficial to elaborate on their potential impact on the study's validity and generalizability. The discussion briefly mentions the need for further investigations but could provide more specific recommendations for future research.
Regrettably, this paper is not written appropriately. I must express my sincere apologies, but it appears that it needs to be rewritten from the beginning, approached with patience and a thorough understanding of the subject matter. I understand the considerable effort and time the authors have invested in the preparation of this manuscript, and it is with reluctance that I must say that it is not acceptable in its current form. My intention is to encourage the authors to consider a comprehensive rewrite, ensuring clarity, coherence, and precision.
Comments on the Quality of English LanguageExtensive editing of English language required
Reviewer 2 Report
Comments and Suggestions for Authors
In the present study, the authors report elevated levels of PEDF, a matricellular protein which is induced also in brain tissue during the course of patients with subarachnoid hemorrhage due to ruptured aneurysms. They included the elderly population and showed a significant increase in patients with poor outcome. The level of this protein is elevated generally in SAH in comparison to patients with unruptured aneurysms, which could also be shown. This is a very interesting and new finding which could help to monitor and predict the outcome of patients with subarachnoid hemorrhage and therefore be helpful in clinical decisions especially when a poor outcome is very likely to be expected.
It would be very interesting to see if these results can be reproduced in a randomized study.
Also it would be of great interest if the levels of PEDF in the younger patients with SAH also allow any predictions concerning the clinical outcome. Moreover, other cerebral pathologies (e.g. intracerebral hemorrhage, gliomas) can also be monitored with this protein.
I therefore recommend this paper for publication.
Round 2
Reviewer 1 Report
Comments and Suggestions for Authors
The authors have answered all my questions. Therefore, I recommend the publication of this Manuscript.